# Photoelectrocatalytic Processes of TiO_2_ Film: The Dominating Factors for the Degradation of Methyl Orange and the Understanding of Mechanism

**DOI:** 10.3390/molecules28247967

**Published:** 2023-12-06

**Authors:** Yuhui Xiong, Sijie Ma, Xiaodong Hong, Jiapeng Long, Guangjin Wang

**Affiliations:** 1School of Materials Science and Engineering, Shenyang University of Chemical Technology, Shenyang 110142, China; 2013020230@stu.syuct.edu.cn; 2School of Materials Science and Hydrogen Energy, Foshan University, Foshan 528051, China; 15729508936@163.com (S.M.); hongxiaodong@lntu.edu.cn (X.H.)

**Keywords:** TiO_2_, photoelectrochemical performance, photoelectrocatalytic performance, methyl orange, reactive oxygen species

## Abstract

Various thicknesses of TiO_2_ films were prepared by the sol–gel method and spin-coating process. These prepared TiO_2_ films exhibit thickness-dependent photoelectrochemical performance. The 1.09-μm-thickTiO_2_ film with 20 spin-coating layers (TiO_2_-20) exhibits the highest short circuit current of 0.21 mAcm^−2^ and open circuit voltage of 0.58 V among all samples and exhibits a low PEC reaction energy barrier and fast kinetic process. Photoelectrocatalytic (PEC) degradation of methyl orange (MO) by TiO_2_ films was carried out under UV light. The roles of bias, film thickness, pH value, and ion properties were systematically studied because they are the four most important factors dominating the PEC performance of TiO_2_ films. The optimized values of bias, film thickness, and pH are 1.0 V, 1.09 μm, and 12, respectively, which were obtained according to the data of the PEC degradation of MO. The effect of ion properties on the PEC efficiency of TiO_2_-20 was also analyzed by using halide as targeted ions. The “activated” halide ions significantly promoted the PEC efficiency and the order was determined as Br > Cl > F. The PEC efficiency increased with increasing Cl content, up until the optimized value of 30 × 10^−3^ M. Finally, a complete degradation of MO by TiO_2_-20 was achieved in 1.5 h, with total optimization of the four factors: 1.0 V bias, 1.09-μm-thick, pH 12, and 30 × 10^−3^ M Cl ion content. The roles of reactive oxygen species and electric charge of photoelectrodes were also explored based on photoelectrochemical characterizations and membrane-separated reactors. Hydrogen peroxide, superoxide radical, and hydroxyl radical were found responsible for the decolorization of MO.

## 1. Introduction

Energy and environmental crises have become serious threats to human society in the world today. Industrialization and population growth are major factors that result in environmental pollution and energy shortages [1]. Organic materials including dyes dumped into the water by organic industries are hazardous to aquatic life. The production of synthetic dyes has increased rapidly to meet the demands of the textile, food, printing, ink, tannery, paper, pharmaceutical, and cosmetic industries. However, the chemicals used in the production of these dyes are toxic and carcinogenic [2]. Therefore, it is obligatory for the scientific community to explore ways for overcoming the issues related to industrial effluents loaded with dye contaminants [3].

Photoelectrocatalysis (PECs) has attracted considerable interest since the discovery of the Honda–Fujishima effect [4] in 1972, especially following the serial stories reported by Carey et al. [5,6] that degradation-resistant organics such as polychlorinated biphenyl can be decomposed by TiO_2_ photoelectrode. In 1989, Tanaka, K. et al. [7] proved that reactive oxygen species (ROSs) such as hydroxyl radicals (•OH) play a key role in photocatalysis, achieving great progress in photocatalytic (PC) mechanism. Furthermore, ROSs can be more effectively produced in photoelectrocatalytic (PEC) processes in a voltage bias photoelectrode, contributing significantly to the PEC synergy effect and enhanced PEC efficiency [8,9]. PECs combines the advantages of both PC and electrocatalytic technologies. Palanisamy’s group [10] demonstrated that the PEC process effectively eliminated 76.2% of amoxicillin within 120 min at 0.8 V, outperforming the removal rates attained by the PC (52.6%) and electrocatalytic (32.3%) processes. PECs can both make use of the solar energy and regulate the photocatalytic process by using photoelectrodes with an appropriate external bias. Moreover, the PEC process is convenient for recycling photoelectrodes, avoiding the dilemma encountered in the photocatalytic process. PECs have been regarded as a technique revolution owing to the combination of photocatalysis, electrocatalysis, and solar energy utilization [11].

Various factors affecting PEC performance have been widely reported for TiO_2_ photoelectrodes. Zhao’s group [12] observed that anatase TiO_2_ with a {001} crystal plane exposed shows prominent PEC activity. Wang and co-authors [13] demonstrated that the PEC efficiency increases when the UV light intensity ranges from 700 μWcm^−2^ to 2.5 mWcm^−2^. Iltaf Shah et al. [2] revealed that a basic alkaline medium is more suitable for a higher degradation rate of ethyl violet dye than acidic and neutral media. Lu Li and his teammates [14] announced that the PEC kinetics constant (K’) was a 6.0-fold increase compared to the PC system. They observed that the PEC degradation efficiency of o-chlorophenol was 96.6% in 180 min under optimum conditions (bias: 0.5 V, solution pH: 6.3). Yan et al. [15] claimed that the PEC efficiency is approximately proportional to the thickness of the TiO_2_ film; however, excessive thickness causes deterioration of the PEC efficiency. Anderson’s team [16] reported the effect of bias, pH, inorganic ions on the PEC efficiency. Particularly, halide ions can be “activated” by a photoelectrode with sufficient bias and transformed into an “active halide” with highly catalytic oxidation activity, greatly improving the PEC efficiency of decomposing seawater to produce hydrogen [17]. To conclude, extensive investigations have been carried out on one or several of the factors affecting the PEC efficiency of TiO_2_, including the crystal phase, light source and light intensity, film thickness, bias, ion properties, and pH value [11]. However, a systematic study on the factors dominating the PEC efficiency and related mechanisms is still absent so far.

Numerous metal oxides such as TiO_2_, SnO_2_, ZnO, WO_3_, and Cu_2_O have been used as photoelectrodes in PECs [18]. TiO_2_ is considered one of the most important photoelectrodes due to its high activity, stability, and low cost [19]. Various forms of nanostructured TiO_2_ film have been used in various applications such as detectors, memories, high-efficiency hydrolysis, diodes, transistors, sensors, etc. [20,21]. With regard to the feasibility of the application of TiO_2_ photoelectrodes, film thickness, bias, ion properties, and pH value are the four most important factors. Therefore, in this study, we initiate a systematic study on the four factors dominating the PEC efficiency. The roles of ROSs and the electric charge of electrodes were also investigated to explore the related PEC mechanisms based on photoelectrochemical characterizations combined with membrane-separated reactors. This study might be beneficial to PEC research fields involved in the appropriate choice and optimization of experimental conditions and parameters.

## 2. Results

### 2.1. Structure, Morphology, and Optimal Properties of TiO_2_ Films

Figure 1a shows the X ray diffraction (XRD) pattern of TiO_2_-20. All the diffraction peaks in the pattern are well-indexed to TiO_2_ and F-doped tin oxide (FTO). The peaks located at 2θ = 25.3°, 37.8°, 39.2°, 48.1°, 55.1°, 56.2°, and 69.1° are attributed to anatase types of TiO_2_. No other diffraction peak arises from possible impurities, indicating that a pure-phase TiO_2_ film is produced. The variation curves of thickness and absorbance of the films are given as the number of layers in Figure 1b. It is obvious that both the thickness and absorbance of the films increase with increasing layers of coatings.

Figure 2a–c shows the scanning electron microscopy (SEM) images of TiO_2_-1, TiO_2_-4, and TiO_2_-20. TiO_2_-1 has a uniform, smooth, and dense surface (Figure 2c). We observed that the surface morphologies of TiO_2_-2 and -3 are similar to that of TiO_2_-1. Cracks began to appear in TiO_2_-4 and then gradually widened as the layers increased. SEM images of TiO_2_-4 and -20 in Figure 2b,c are given, respectively, as representative examples. The inset in Figure 2a is the high-magnification SEM image of TiO_2_-1. The surface morphology in the red box in Figure 2b,c is similar to that of the inset. The results indicate that all TiO_2_ films are composed of nanoparticles with a size of 20–30 nm. The microstructure of the TiO_2_ films was further investigated with transmission electron microscopy (TEM). Figure 2d shows the representative TEM images of TiO_2_-20. Grain crystal particles of TiO_2_-20 shown in the TEM image can be clearly observed, which also indicate that TiO_2_-20 is composed of 20−30 nm size nanoparticles.

### 2.2. Photoelectrochemical Properties of TiO_2_ Photoelectrodes

Before the measurement of PEC regulation of MO by TiO_2_ photoelectrodes, the cyclic voltammetry (CV) technique was used to examine the charge transfer between the FTO and the electrolyte. Figure 3a shows that the onset potentials for the anodic and cathodic dark current are essentially independent of the film thickness and scan rate. Representative samples of TiO_2_-1, -8, -20, and -32 displayed nearly perfect blocking characteristics, indicating that all the films are pinhole free [22]. The result is also supported by the SEM data showing that the surface of the FTO substrate is completely covered by a very thin and compact TiO_2_ coating. Moreover, even if there are cracks in some of the TiO_2_ samples such as TiO_2_-20, the inner coatings contacting the FTO substrates are still uniform and dense enough to separate FTO substrates from electrolytes, blocking charge transfer between them. In this case, charge transfer can only proceed between the TiO_2_ photoelectrode and the electrolyte in the PEC reaction. The position of the flat-band potential was obtained by taking the x intercept of the Mott–Schottky plots of the TiO_2_ films (Figure 3b), which gave all the samples a considerably negative flat-band potential, for example, −0.51 V for TiO_2_-20. The positive slope of the plots again proved the n-type nature of the TiO_2_ films. The application of a potential over the flat-band potential can suppress the charge carrier recombination. Thus, a zero or positive bias over the flat-band potential on the TiO_2_ film electrodes may improve the PEC efficiency. 

Figure 4a,b shows the variation in the short circuit current (I_SC_) and open circuit voltage (V_OC_) as the layers of TiO_2_ coatings. The data of I_SC_, V_OC_, and film thickness were abstracted from Appendix A. The behavior of the anode photocurrent and shift to negative potential under irradiation in Appendix A indicate an n-type semiconductor behavior. The I_SC_ and V_OC_ increased at first and then decreased as the film thickness increased. TiO_2_-20 achieved the highest I_SC_ of 0.21 mAcm^−2^ and V_OC_ of 0.58 V. The thickness-dependent photoelectrochemical properties of the TiO_2_ films were also observed by M. L. Hitchman [23] and M. Rodríguez-Pérez [24] in polycrystalline anatase. EIS is particularly useful for explaining the interface charge transport and recombination [25]. Figure 4c shows Nyquist plots of the TiO_2_-1, -8, -16, -20, -24, -32. The zoom of the high-frequency range in Figure 4d indicates that the semicircle part of TiO_2_-20 has a minimum diameter among all the TiO_2_ films. In the same frequency, a small impedance semicircle indicates a large capacitance value at the corresponding time constant and a low Faraday current impedance, signifying a fast kinetic process [26]. Thus, TiO_2_-20 was chosen and used for the follow-up PEC experiment owing to its fast kinetic process and low PEC reaction energy barrier.

### 2.3. PEC Degradation of MO by TiO_2_ Photoelectrodes

We carried out the PEC degradation of MO using the PEC reactor as shown in Figure 1a. Figure 5a shows the variation in the degradation rate of pristine MO using the TiO_2_-20 photoelectrode, with the potential bias ranging from 0 V to 1.2 V. Generally, in the PEC process, the degradation rate increased with an increasing bias, up until the optimized value of 1.0 V. Song et al. [27] and Zanoni et al. [28] reported similar results, in which at the optimal potential bias (for a given light intensity and film thickness), the electrons and holes were so well separated that enhancing the potential bias led to no significant improvement in the PEC activity. The PEC degradation of MO was further carried out using different thicknesses of TiO_2_ photoelectrodes. As expected, TiO_2_-20 achieved the highest degradation rate of 88.6% in 6 h among the representative samples at a bias of 1.0 V (Appendix A). This result is in accordance with the data of photoelectrochemical characterizations. Therefore, PEC regulation of MO by TiO_2_-20 with a 1.0 V bias was chosen and used for the tests of different pH, halide ion, and concentrations of Cl, respectively.

Figure 5b shows the degradation rates of MO by TiO_2_-20 with 1.0 V bias at different pH levels of 1, 4, 7, 9, and 12. The pH value of the solution was adjusted by dropwise adding H_2_SO_4_ and NaOH. The pH value of original MO solution in the control experiment was 5.8. One can see that the increasing pH value contributes to an enhanced degradation rate. Moreover, the degradation rates of all pH-adjusted MO solutions are higher than that of the pristine one. The reason is that the high conductivity of the MO solution increases the mass transfer and improves the PEC efficiency due to the addition of electrolytes H_2_SO_4_ and/or NaOH [29].

The influence of different electrolytes on the PEC degradation of organics has extensively been investigated in NaCl, NaNO_3_ and Na_2_SO_4_, NaBr, and NaClO_4_ [15,29,30]. The effects depend on the types and concentrations of negatively charged ions [11]. Halide ions can be transformed into “active halide” with highly catalytic oxidation activity, substantially improving the PEC efficiency and producing organic halides of industrial and medicinal importance [17]. Thus, in this study, we focused on the effect of halide ions on the PEC efficiency of TiO_2_-20. Iodine ion (I^−^) was not included here because of its highly chemical reduction [31]. Figure 5c shows the degradation of MO by halide ions F, Cl, and Br. The order of the PEC efficiency was determined as Br > Cl > F. The halide ions can be activated to form highly oxidizing free radicals, significantly improving the PEC efficiency [30]. Considering that the Cl ion is very important and spreads over the water on earth, we carried out PEC experiments on the degradation of MO with different concentrations of Cl ion. Figure 5d shows that the addition of Cl greatly increased the photocurrent and PEC efficiency. The degradation rate rapidly increased as the Cl content increased from 0 to 10 × 10^−3^ M. However, the increase in the degradation rate slowed down as the concentration of Cl ranged from 10 × 10^−3^ M to 30 × 10^−3^ M. Moreover, excessive Cl content caused a decrease in degradation rate. The results indicate that the content of “active” Cl reached nearly its maximum, meaning that increasing the content of Cl cannot produce more “active” Cl but suppresses the PEC efficiency. In all, the PEC efficiency can be substantially improved by optimizing factors dominating the PEC performance of TiO_2_ films. A complete degradation of MO by TiO_2_-20 was achieved in 1.5 h in the PEC process with total optimization of the four factors: a thickness of 1.09 μm, a bias of 1.0 V, a content of 30 × 10^−3^ M Cl ion, and pH 12. The PEC degradation efficiency of MO by TiO_2_-20 is pretty high and typical examples of comparative literature values are included in Table 1.

### 2.4. The Contrast of PEC and PC Degradation of MO

The contrast of the PEC and PC efficiency of TiO_2_-20 was carried out using PCMS and PECMS reactors, as shown in Figure 1c,d. Figure 6a shows that the degradation rate of MO by TiO_2_-20 in the PCMS reactor is 41.5% in 6 h in the PC process, while that with a 1.0 bias in the PCEMS reactor is 100% in 5 h in the PEC process. Figure 6b shows that the TiO_2_-20 in the PC reactor earned 26.3% degradation of MO, while the same TiO_2_-20 with 0 V bias using the PEC reactor gained an enhanced degradation rate of 32.3%. It is expected that the application of a potential (0 V) over the flat-band potential (−0.51 V for TiO_2_-20) can suppress the charge carrier recombination and improve the PEC efficiency according to data of photoelectrochemical characterizations. These results indicate that the PEC process under electrochemical control achieves a much higher degradation rate than the PC one.

A comparison of the PEC efficiency of TiO_2_-20 in a PEC reactor with the PC efficiency of TiO_2_ powder in a PC reactor was also carried out. The film mass of TiO_2_-20 was determined as 3.81 mg by measuring the difference of mass between FTO and TiO_2_-20-coated FTO. Figure 6b shows that the single TiO_2_-20 film can decolor 26.3% of MO, while TiO_2_ powder (3.81 mg) can decompose 44.3% of MO in 6 h. The reason is that the TiO_2_ powder is highly dispersed and has much larger contact area with MO solution than that of the single TiO_2_-20 film. However, the TiO_2_ film photoelectrode is very easy to be recycled and reused compared to the TiO_2_ powder catalyst. Moreover, the TiO_2_ film photoelectrode is convenient for the PEC process. Particularly, the PEC efficiency can be improved by electrochemical regulation; for example, TiO_2_-20 with a bias of 0.5 V can achieve an equivalent degradation rate of MO as a 3.81 mg TiO_2_ powder (Figure 6b). 

### 2.5. The Roles of ROSs and Electric Charge of Electrodes

The general principle of PECs is well-documented in the pioneering work and follow-up studies [4,37,38,39,40,41,42]. Figure 2 shows the energy level diagram of the TiO_2_ film and the events possibly occurring during the PEC process. Under UV irradiation, the electrons (e^−^) in TiO_2_ are excited to the conduction band, which creates holes (h^+^) in the valence band. The excited holes can oxidize water into oxygen and interact with O_2_ and H_2_O to form an assortment of ROSs, (i.e., •OH, H_2_O_2_). The excited electrons are collected and removed by an external circuit to the counter electrode, where they are frequently captured by oxygen to form ROSs (i.e., •O_2_^−^, H_2_O_2_). Here, we carried out PC and PEC tests to explore the origin of ROSs with a partition system (see Figure 1c,d) under different conditions.

Figure 7a shows that the •OH and•O_2_^−^ yield increased with increased irradiation time in both TiO_2_ and Pt cells of the PCEMS reactor (Figure 1d). The species of •OH in the TiO_2_ of the PCEMS reactor were probably produced by a hole-induced oxidation process [43]:H_2_O + h^+^→•OH + H^+^(1)

The •OH yield in the Pt cell of the PCEMS reactor should originate from the Pt electrode because •OH in TiO_2_ cell has a very short lifetime and cannot pass through the membrane to the Pt cell. It well known that electrons can react with O_2_ to produce •O_2_^−^, and then H_2_O_2_ and •OH through a reductive process (Formulas (2)–(6)) [44,45]. The very low •OH yield suggests that a low level of kinetics evolution of •OH proceeds in the Pt cell of a PCEMS reactor.
(2)$$O_{2}+e^{-}\to •{O_{2}}^{-}\overset{{2H}^{+}}{\to }H_{2}O_{2}$$
O_2_ + 2H_2_O + 2 e^−^→H_2_O_2_ + 2OH^−^(3)
H_2_O_2_ +•O_2_^−^→•OH + OH^−^+ O_2_
(4)
2•OH→H_2_O_2_
(5)
(6)$$H_{2}O_{2}\overset{hv}{\to }2•OH$$

Figure 7b shows that all of the H_2_O_2_ yields increased at first and became stable with prolonged irradiation time due to the decomposition of H_2_O_2_ in parallel with its production. The equilibrium concentration of H_2_O_2_ in the TiO_2_ and blank cells of the PEMS reactor was estimated to be 1.48 and 4.8 μM, respectively. It is no doubt that no PC process happened in the blank cell of the PEMS reactor so the supply of H_2_O_2_ in the blank cell attributed to the diffusion of H_2_O_2_, which was produced in the PC process of the TiO_2_ cell in the PEMS reactor and had a long life span and could pass through the membrane [46].

The equilibrium concentration for H_2_O_2_ in the TiO_2_ and Pt cells of the PECMS reactor significantly increased up to 4.58 and 3.5 μM, respectively. Although H_2_O_2_ can be derived from •OH according to Formulas (3) and (5), the formation kinetics reaction of H_2_O_2_ is very slow due to its second-order reaction. In this case, the much higher H_2_O_2_ yield in the TiO_2_ cell compared to the Pt cell of the PECMS reactor suggests that H_2_O_2_ can be probably produced from other sources. We believe that the supply of H_2_O_2_ in the TiO_2_ cell of the PECMS reactor originates from the back electrons (see the red arrow in Figure 2) of TiO_2_ [47], which were transferred to the solution and underwent a reductive process as Formulas (2)–(5). The related mechanism and detailed descriptions on back electrons will be published elsewhere. 

Note that an 8.5% degradation of MO was observed in 6 h in the blank cell of the PCMS and little degradation of MO under light control indicates that UV light cannot decompose MO in our experimental conditions (Figure 6a). In addition, ROSs including •OH and•O_2_^−^ were produced in the TiO_2_ cell of the PCMS, but they cannot pass through the semi-membrane because of their short lifetime. In this case, only H_2_O_2_ is responsible for the 8.5% degradation of MO in the blank cell of the PCEMS. It is reported that •O_2_^−^ with highly catalytic oxidation activity can also decompose MO [48]. So, H_2_O_2_, •O_2_^−^, and •OH were responsible for the decolorization of MO. The much higher yields of H_2_O_2_, •O_2_^−^, and •OH in the TiO_2_ cell than those in the Pt cell of the PCEMS reactor suggest that the TiO_2_ photoelectrode plays a major role in the PEC process.

## 3. Materials and Methods

### 3.1. Materials

Tetrabutyl titanate (99% purity), sodium sulfate (99.0% purity), absolute ethanol (99.9% purity), sodium fluoride, sodium chloride, sodium bromide, MO, sulfuric acid, sodium hydroxyl, and nitric acid were purchased from Sinopharm Group (Shanghai, China). 2,3-bis(2-methoxy-4-nitro-5-sulfophehyl)-2H-tetrazolium-5-carboxanilide (XTT) (>98% purity), Terephthalic acid (TA) (98% purity), and horseradish peroxidase (POD) (activity: 250–330 units/mg solid) were received from Sigma-Aldrich (Shanghai, China). FTO glass was obtained from Youxuan Technology Corp. (Liaoning, China). All were used as obtained. The solutions used in this work were prepared with deionized water further purified with a Millipore Milli-Q (Millipore, Bedford, MA, USA) purification system (resistivity 18.6 MΩ).

### 3.2. The Preparation of TiO_2_ Film Photoelectrodes and TiO_2_ Powder

In this step, 15 mL ethanol containing 0.3 mL water was added dropwise into the mixture of 5 mL tetrabutyl titanate, 30 mL ethanol, and 0.3 mL nitric acid, stirring vigorously. Then, the whole mixture was continuously stirred for 2 h. A colorless and transparent TiO_2_gel was obtained by keeping the whole mixture sealed for one day. On an FTO glass substrate (1.5 × 2.5 cm^2^), a thin film of sol was spin-coated at 2000 rpm for 30 s, followed by annealing in air with different ramping rates to the final 500 °C. FTO substrates with 1, 2, 4, 8, 16, 20, 24, 32 layers of coatings were achieved by repeating the spin-coating process and were denoted as TiO_2_-1, -2, -4, -8, -16, -20, -24, and -32, respectively. TiO_2_ electrodes with 1 × 1 cm^2^ of surface exposed were prepared by sealing them with epoxy resin. TiO_2_-20 was chosen for XRD measurement. TiO_2_ powder was collected by scraping the films off several TiO_2_-20 samples.

### 3.3. Characterization of TiO_2_ Films

XRD measurements were carried out on a PW 1840 powder X-ray diffractometer, using Cu Ka (1.54Å) as the incident radiation. SEM images were obtained on a field-emission scanning electron microscopy (JSM-6700F, JEOL, Tokyo, Japan) at 30 kV. TEM were carried out with JEOL JEM100CXII. The absorption spectra were measured by a UV–vis spectrophotometer (CARY5000, Varian, Australia). The thickness of films was determined by a step profiler (Dektak XT, Bruker, Germany). Photoelectrochemical measurements and characterizations were conducted by a three-electrode system with a TiO_2_ film electrode as working electrode, Pt plate electrode as counter electrode, and Ag/AgCl electrode as reference electrode, respectively, on a CHI660D station (Chenhua, Shanghai, China). The electrolytes were either 0.5 × 10^−3^ M K_4_Fe(CN)_6_ + 0.5 × 10^−3^ M K_3_Fe(CN)_6_ or air-saturated 1.0 × 10^−3^ M Na_2_SO_4_ solution. The light source was a UV-LED whose spectrum was given in Appendix A. The light intensity was measured by a light meter (LI-COR, Lincoln, NE, USA), and the light intensity for the experiments was fixed at 100 mW/cm^2^.

### 3.4. Detection of •OH,•O_2_^−^ and H_2_O_2_

The production of •OH was detected by a photoluminescence (PL) method by using terephthalic acid (TA) as a probe molecule [49]. The experimental procedure was similar to the measurement of PEC and PC activity except that the MO aqueous solution was replaced by the 5 × 10^–4^ M TA aqueous solution with a concentration of 2 × 10^–3^ M NaOH. The superoxide radical (•O_2_^−^ was measured by XTT [50,51], which can be reduced by •O_2_^−^ to form XTT-formazan. The formazan has an absorption spectrum (measured by UV/Vis spectrophotometer (Blue Star A, Fort Lauderdale, FL, USA) with a peak at 470 nm can be used to quantify the relative amount of superoxide. H_2_O_2_ was analyzed photometrically by the Peroxidase (POD)-catalyzed oxidation product of DPD [52,53], which was measured by UV/Vis spectrophotometer (CARY5000, Varian, Australia) at 551 nm.

### 3.5. PC and PEC Experiments of TiO_2_ Photoelectrodes

Figure 1 shows four kinds of reactors: (a) PEC reactor, in which TiO_2_ film, Pt were used as working electrode, counter electrode, respectively; (b) PC reactor, in which TiO_2_ film was dipped in solution; (c) membrane-separated (MS) reactor, and (d) PEC membrane-separated (PECMS) reactors which are the same as (a), (b), respectively, except that the reactors are separated into two compartments by a semipermeable membrane. The photographs of experimental set-up of PEC, PC reactors and MS, PECMS reactors are given in Appendix A. The pore size of the semipermeable membrane was 5 nm, whereas that of MO was about 6–8 nm in size so that MO cannot pass through the membrane. PEC degradation of MO was conducted with the TiO_2_film as working electrode, Pt wire electrode as counter electrode, respectively, on a DXW-12V100A DC Voltage Regulator (Suzhou, China). At different time intervals, aliquots of the sample were collected. The MO concentration was analyzed by recording variations in the absorption band maximum at 465 nm (defined as A_t_) in the UV-vis spectra of MO by using a UV-vis spectrophotometer. MO concentration of the reaction solution was defined as A_0′_. The degradation efficiency of MO was calculated according to the equation: degradation rate (%) = (A_0′_ – A_t_)/A_0′_ × 100%. The liquid phase degradation of MO was used for the evaluation of the PC activity of the TiO_2_ powder.

## 4. Conclusions

The effects of bias, film thickness, pH value, and ion properties on the PEC performance of TiO_2_ films were systematically studied under UV irradiation. At an optimized bias of 1.0 V, the TiO_2_-20 photoelectrode can degrade 84.5% of MO in 6 h, which outperforms the other TiO_2_ film samples. We observed that a high pH value contributed to enhanced degradation of MO. The “activated” halide ions can significantly promote PEC efficiency and the order of PEC efficiency was determined as Br > Cl > F. The degradation rate increased with an increasing Cl content in an MO solution, up until the optimized value of 30 × 10^−3^ M. However, excessive Cl content causes a decrease in degradation rate. The PEC efficiency can be significantly improved and a complete degradation of MO was achieved in 1.5 h using TiO_2_-20 with 1.0 V bias and 30 × 10^−3^ M Cl ion content at Ph 12. The roles of ROSs and electric charge of electrodes were investigated to explore the related PEC mechanisms and H_2_O_2_, •O_2_^−^, and •OH were found responsible for the decolorization of MO.

## Data Availability

Data are contained within the article and Appendix A.

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
