# Peer review of "Photoelectrocatalytic Processes of TiO_2_ Film: The Dominating Factors for the Degradation of Methyl Orange and the Understanding of Mechanism"

_molecules, 2023, doi:10.3390/molecules28247967_

Round 1

Reviewer 1 Report

Comments and Suggestions for Authors

Yuhui Xiong et al. reported “Photoelectrocatalytic processes of TiO2 film: the dominating factors for the degradation of methyl orange and the under-standing of mechanism”. The work is publishable after addressing the following issues.

1.       Remove the abbreviation from abstract part.

2.       The abstract writing is very poor. Should be re-write with English expert.

3.       The introduction section need extensive revision. Here I have not seen any part about dyes and its effective on environment. Read and cite the following literature, which will be helpful to improve the introduction section and will be useful for the readers. ACS Omega 2022, 7, 3415434165 (doi.org/10.1021/acsomega.2c03472), RSC Adv., 2022, 12, 15658–15669 (DOI: 10.1039/d2ra01722a).

4.       In the introduction section before Photoelectrocatalysis, include other methods used for the removal or degradation of organic pollutants and why Photoelectrocatalysis is preferred over others. Although we know that Photoelectrocatalysis is expensive and energy consuming method.  

5.       SEM is useful when someone need the surface morphology in micrometer range and not so useful in nm range. For the present study, TEM should be included in the revised manuscript, which will confirm the surface morphology more in depth.

6.       BET analysis should be included in the revised manuscript because surface area is a fundamental factor in the present study which effect the performance of photocatalyst.

7.       The mechanism of dye degradation should be included in the revised manuscript in the form of sketch.

8.       Comparative table with other literature values should be included in the revised manuscript.

9.       Improve the English via English expert because some sections are written very poorly.

10.   Check the spelling and grammatical mistakes throughout the manuscript. 

Comments on the Quality of English Language

Moderate editing of English language required

Author Response

Yuhui Xiong et al. reported “Photoelectrocatalytic processes of TiO2 film: the dominating factors for the degradation of methyl orange and the under-standing of mechanism”. The work is publishable after addressing the following issues.

  1. Remove the abbreviation from abstract part.

Response (R): Thank reviewer’s kind suggestion. We looked up the information of format requirement for the abstract on website of MDPI Style Guide (https://www.mdpi.com/authors/layout#_bookmark5). It claimed that any abbreviations used must be defined in the abstract. We double checked all abbreviations in the abstract and we confirmed all of the abbreviations were defined previously in our original manuscript. Thus, we retained the abbreviations in the abstract in the revision in order to keep the text of the abstract concise and clear. We wish the reviewer’s kind understanding.

  1. The abstract writing is very poor. Should be re-write with English expert.

R: We appreciate the reviewer’ suggestion. We double checked the abstract writing and polished carefully it. Now, we believe that the abstract is in a good shape.

  1. The introduction section need extensive revision. Here I have not seen any part about dyes and its effective on environment. Read and cite the following literature, which will be helpful to improve the introduction section and will be useful for the readers. ACS Omega 2022, 7, 34154−34165 (doi.org/10.1021/acsomega.2c03472), RSC Adv., 2022, 12, 15658–15669 (DOI: 10.1039/d2ra01722a).

R: Thank the reviewer’s kind suggestion. We read carefully the literatures (ACS Omega 2022, 7, 34154−34165, RSC Adv., 2022, 12, 15658–15669) that the referee recommended. We think the literatures are helpful for the introduction section and will be useful for the readers. The literatures were cited in our revision as references 2 and 3 in the introduction section.

  1. In the introduction section before Photoelectrocatalysis, include other methods used for the removal or degradation of organic pollutants and why Photoelectrocatalysis is preferred over others. Although we know that Photoelectrocatalysis is expensive and energy consuming method.

R: Yes, we included some more methods such as photocatalysis and electrocatalysis for the removal or degradation of organic pollutants and why photoelectrocatalysis is preferred over others in the revised manuscript. For example, we cited reference 10 in which Palanisamy et al. demonstrated that the PEC process effectively eliminated 76.2% of amoxicillin within 120 min at 0.8 V, outperforming the removal rates attained by the PC (52.6%) and electrocatalytic process (32.3%). Furthermore, we clarified the advantage over others. See lines 56-59 in the Introduction section of the revised manuscript: “PECs can both take make use of the solar energy and regulate the photocatalytic process by using photoelectrodes with an appropriate external bias. Moreover, the pho-to-electrodes PEC are convenient for recycling in PEC processes, avoiding the dilemma encountered in the photocatalytic process.”

  1. SEM is useful when someone need the surface morphology in micrometer range and not so useful in nm range. For the present study, TEM should be included in the revised manuscript, which will confirm the surface morphology more in depth.

R: Yes, we did so. The TEM image of TiO2 was added in the revision as the referee’ suggestion.

  1. BET analysis should be included in the revised manuscript because surface area is a fundamental factor in the present study which effect the performance of photocatalyst.

R: We agree that BET data is a fundamental factor that effects the performance of photocatalyst. We appreciate the reviewer’s valuable comments. It well known that BET is frequently carried out to measure the specific surface area of a powder sample. However, in our study, we used a TiO2 film/FTO photoelectrode to the photoelectrocatalytic degradation of methyl orange. TiO2 film coated on FTO is inconvenient to conduct a BET measurement. We appreciate the reviewer’ kind consideration.

  1. The mechanism of dye degradation should be included in the revised manuscript in the form of sketch.

R: We put a schematic diagram in the revised manuscript according to the reviewer’s nice suggestion. Please see Sketch 2 in the revision: Energy level diagram of TiO2 film and the events possibly occurring during the PEC process.

  1. Comparative table with other literature values should be included in the revised manuscript.

R: We took some examples of TiO2 electrodes for dye degradation using PEC systems, which are included in Table 1 in the revised manuscript.

  1. Improve the English via English expert because some sections are written very poorly.

R: Thank reviewer’s kind suggestion. We double checked whole manuscript and carefully polished it. Now we believe that the revision is in good shape.

  1. Check the spelling and grammatical mistakes throughout the manuscript.thoroughly spell-check

R: Thank reviewer’s kind suggestion. We double checked the spelling and grammatical mistakes throughout the manuscript.

Reviewer 2 Report

Comments and Suggestions for Authors

Although this is a rather interesting article that can be recommended for publication, however some comments/questions should be taken in to account.

1.     Introduction. More information about TiO2, its nanostructures, what has been done in recent years should be updated. It is also important from the point of view that the article should be visible to a wide range of readers. It is important to note that there are several modification of TiO2 and they give different effects.

See, some of them, recently published in MDPI:

Tsebriienko, T.; Popov, A.I. Effect of Poly(Titanium Oxide) on the Viscoelastic and Thermophysical Properties of Interpenetrating Polymer Networks. Crystals 202111, 794. https://doi.org/10.3390/cryst11070794

Ge, S., Sang, D., Zou, L., Yao, Y., Zhou, C., Fu, H., & Wang, C. (2023). A Review on the Progress of Optoelectronic Devices Based on TiO2 Thin Films and Nanomaterials. Nanomaterials13(7), 1141.

Jiao, M., Zhao, X., He, X., Wang, G., Zhang, W., Rong, Q., & Nguyen, D. (2023). High-performance MEMS oxygen sensors based on Au/TiO2 films. Chemosensors11(9), 476.

2.     Most of the references in the Introduction are quite old and do not reflect current knowledge of the literature.

3.     Line 63-64.  Is it possible to formulate a “figure of merit”, using these 4 factors?

4.     Figure 4 (a, b).  These data need error bars.

5.     Figure 7 and corresponding data in the text.   Exposure time, expressed in hours, is not an absolute value, since this time depends on the corresponding exposure source, the corresponding distance, optics, etc.

In general, the manuscript is interesting and can be recommended for publication after constructive reflection on the above comments.

Author Response

Although this is a rather interesting article that can be recommended for publication, however some comments/questions should be taken in to account.

  1. Introduction. More information about TiO2, its nanostructures, what has been done in recent years should be updated. It is also important from the point of view that the article should be visible to a wide range of readers. It is important to note that there are several modification of TiO2 and they give different effects.

See, some of them, recently published in MDPI:

Tsebriienko, T.; Popov, A.I. Effect of Poly(Titanium Oxide) on the Viscoelastic and Thermophysical Properties of Interpenetrating Polymer Networks. Crystals 2021, 11, 794. https://doi.org/10.3390/cryst11070794

Ge, S., Sang, D., Zou, L., Yao, Y., Zhou, C., Fu, H., & Wang, C. (2023). A Review on the Progress of Optoelectronic Devices Based on TiO2 Thin Films and Nanomaterials. Nanomaterials, 13(7), 1141.

Jiao, M., Zhao, X., He, X., Wang, G., Zhang, W., Rong, Q., & Nguyen, D. (2023). High-performance MEMS oxygen sensors based on Au/TiO2 films. Chemosensors, 11(9), 476.

R: We read carefully these papers published recently in MDPI. We cited the latter two papers as references 20 and 21 in revised manuscript according to the reviewer’s kind suggestion. However, we are not quite sure that it is appropriate to cite the first paper (Crystals 2021, 11, 794.) because there is unconspicuous relation between the paper and our work. In addition, we also updated some references related to PEC performance of TiO2 photoelectrode. We believe that the Introduction section of the revised manuscript is in a good shape and can be visible to a wide range of readers.

  1. Most of the references in the Introduction are quite old and do not reflect current knowledge of the literature.

R: We updated the references and added refs. 1-3, 10, 14, 20 and 21 into the Introduction of the revised manuscript. We think the references in the Introduction can reflect current knowledge of the literature in current form.

  1. Line 63-64. Is it possible to formulate a “figure of merit”, using these 4 factors?

R: We think it is a good question.“Figure of merit” is a very general concept used for describing the performance of a commodity. It’s one of important indicators to evaluate the quality of a product. In our work, bias, film thickness, pH value and ion properties are four most important factors dominating the PEC performance of TiO2 films. Thus, the four factors don't belong to the indicators describing the performance of the TiO2 film photoelectrode. Therefore, we are not sure that “figure of merit” can be used to formulate these 4 factors. We wish that the reviewer can understand it.

  1. Figure 4 (a, b). These data need error bars.

R: We appreciate reviewer’s kind suggestion. One can see that Figure 4 shows the patterns of variation of short circuit current and open circuit voltage as the layers of coatings of TiO2 films. We acknowledge that random error exists in any experiments. If one wants to obtain the error in the experiments, he has to perform the experiments at least in triplicates. We did so in our previous work in which we carried out all antibacterial experiments in triplicates (LB Xiong et al., Electrochimica Acta 153 (2015) 583–593) because random error frequently exists in biological experiments. Then, we can plot the curves of figures with error bars. However, we are not sure it is necessary to obtain the error data from our experiments. Thus, we keep the Figure 4 (a, b) unchanged and wish our reviewer is considerate of it.

  1. Figure 7 and corresponding data in the text. Exposure time, expressed in hours, is not an absolute value, since this time depends on the corresponding exposure source, the corresponding distance, optics, etc.

R: The reviewer claimed that exposure time expressed in hours in Figure 7 is not an absolute value since the time depends on the corresponding exposure source, the corresponding distance, optics, etc. We are not sure what the reviewer really means. We think that the time is independent of the experimental control conditions such as corresponding exposure source, the corresponding distance, optics, etc. We think the data of Figure 7 is tenable because it is obtained according to the record of data from the test instruments. We wish the reviewer’s kind understanding. 

Round 2

Reviewer 1 Report

Comments and Suggestions for Authors

The author address all the issues in the revised manuscript and now I recommend the manuscript for publication in the present form.

Comments on the Quality of English Language

Ok